# A Combined Differential Proteome and Transcriptome Profiling of Fast- and Slow-Twitch Skeletal Muscle in Pigs

**DOI:** 10.3390/foods11182842

**Published:** 2022-09-14

**Authors:** Wei Wei, Chengwan Zha, Aiwen Jiang, Zhe Chao, Liming Hou, Honglin Liu, Ruihua Huang, Wangjun Wu

**Affiliations:** 1Department of Animal Genetics, Breeding and Reproduction, College of Animal Science and Technology, Nanjing Agricultural University, Nanjing 210095, China; 2Institute of Animal Science and Veterinary Medicine, Hainan Academy of Agricultural Sciences, Haikou 571100, China

**Keywords:** pig, proteome, transcriptome, muscle fiber, meat quality

## Abstract

Skeletal muscle fiber types can contribute in part to affecting pork quality parameters. *Biceps femoris* (Bf) (fast muscle or white muscle) and *Soleus* (Sol) (slow muscle or red muscle) are two typical skeletal muscles characterized by obvious muscle fiber type differences in pigs. However, the critical proteins and potential regulatory mechanisms regulating porcine skeletal muscle fibers have yet to be clearly defined. In this study, the isobaric Tag for Relative and Absolute Quantification (iTRAQ)-based proteome was used to identify the key proteins affecting the skeletal muscle fiber types with Bf and Sol, by integrating the previous transcriptome data, while function enrichment analysis and a protein–protein interaction (PPI) network were utilized to explore the potential regulatory mechanisms of skeletal muscle fibers. A total of 126 differentially abundant proteins (DAPs) between the Bf and Sol were identified, and 12 genes were found to be overlapping between differentially expressed genes (DEGs) and DAPs, which are the critical proteins regulating the formation of skeletal muscle fibers. Functional enrichment and PPI analysis showed that the DAPs were mainly involved in the skeletal-muscle-associated structural proteins, mitochondria and energy metabolism, tricarboxylic acid cycle, fatty acid metabolism, and kinase activity, suggesting that PPI networks including DAPs are the main regulatory network affecting muscle fiber formation. Overall, these data provide valuable information for understanding the molecular mechanism underlying the formation and conversion of muscle fiber types, and provide potential markers for the evaluation of meat quality.

## 1. Introduction

Muscle, one of the most important components of the mammalian body, is composed of various types of muscle fibers, with different physiological and metabolic properties [1] to meet different physiological needs. Meat is mainly produced from muscles derived from slaughtered animals. Meat eating quality is often assessed by various attributes in livestock, such as tenderness, juiciness, and flavor [2], and these meat quality attributes are mainly influenced by the muscle structure and properties [3], whereas the differences in the muscle structure and properties are associated with the muscle fiber types, which are closely related to many meat qualities, such as pH value, intramuscular fat, meat color, water holding capacity, and tenderness [4,5]. Thus, the mechanisms underlying muscle fiber formation are critical for elucidating the molecular mechanism of meat quality in livestock Moreover, identifying the key genes is the key step to systematically elucidate the regulatory mechanism of muscle fiber formation.

With the development of various omics approaches, many genes affecting muscle fiber types in livestock have been identified using transcriptome technology [6,7,8]. However, an inconsistency exists between the transcription and translation, so it is intriguing to identify the key genes regulating the skeletal muscle fiber types at protein level. In recent years, several studies have investigated the proteomic differences in the process of postmortem meat aging [9,10,11], mainly focusing on the dynamic changes that take place in the proteome in the conversion of muscle to meat.

Proteomics is a high-throughput technique used for the quantitative analysis of the entirety proteins in a cell, tissue, or organism under a specific, defined set of conditions. In the past two decades, proteomic technologies have developed rapidly, and the methods are generally classified into two categories: gel-based methods, and gel-free methods. The gel-based methods include one-dimensional electrophoresis (1-DE) and two-dimensional electrophoresis (2-DE), while gel-free techniques include isobaric tag for relative and absolute quantification (iTRAQ), liquid chromatography-electrospray ionization-tandem mass spectrometry (LC-ESI-MS), liquid extraction surface analysis mass spectrometry (LESA-MS), matrix-assisted laser desorption/ionization-time-of-flight (MALDI-TOF), and nano-liquid chromatography-tandem mass spectrometry (nLC-MS/MS); and these proteomic technologies have been widely used to identify the critical proteins associated with meat qualities [9,12,13]. Moreover, the proteomic profiling of various skeletal muscles has been conducted in many species, including cattle [9], sheep [14], horse [15], Chinese perch [16], and mice [17], using different proteomics technologies. Notably, most of these studies conducted the proteomic profiling using meat chilled for a long time, not fresh muscle excised immediately after slaughter. In 2010, Mach et al., compared the porcine proteomic profiling of meat *semimembranosus* and *longissimus dorsi* muscles collected after 24 h of carcass chilling from five different breeds [18], mainly focusing on the dynamic changes of proteins in the postmortem meat aging process using surface-enhanced laser desorption/ionization time-of-flight proteomics technology. From these studies, it can be seen that it is more sensible to investigate proteomic differences of fresh muscles, with various muscle fiber types collected immediately after the slaughter of livestock, which will deepen our understanding of the genes controlling meat quality traits. In addition, the iTRAQ-based proteomic method, as a mainstream technology, has recently been utilized to identify a set of differentially expressed proteins among various groups in specific conditions in many species, such as pigs [19,20], chickens [21], and ducks [22]. The difference in skeletal muscle fiber types is one of the important factors affecting pork quality. *Biceps femoris* (Bf) (fast muscle or white muscle) and *Soleus* (Sol) (slow muscle or red muscle) are two typical skeletal muscles characterized by obvious muscle fiber type differences in pigs; thus, we previously explored their different profiling through the high-throughput whole-transcriptome technique [6]. However, the application of iTRAQ to identify the differential proteomic profiles among porcine skeletal muscles with various muscle fiber types has not been reported yet. Therefore, the objective of this study was to compare the proteomic differences between the porcine fast-twitch muscle Bf and slow-twitch muscle Sol, collected immediately after slaughter using iTRAQ technology, and reveal the key proteins and pathways controlling meat quality traits in pigs.

## 2. Materials and Methods

### 2.1. Animals and Samples Collection

Three full-sibling Duroc × Meishan female pigs numbered 28, 35, and 36 with similar performances were selected for the dissection of skeletal muscle tissues; the pigs were derived from the offspring of a Duroc boar crossed with eight Meishan sows. Two types of skeletal muscles, *Biceps femoris* (Bf) (fast muscle or white muscle) and *Soleus* (Sol) (slow muscle or red muscle), characterized by obvious differences in muscle fiber types, were collected immediately from the same pig after slaughter. Thus, a total of six muscles—namely, Bf28, Bf35, Bf36, Sol28, Sol35, and Sol36—were used for iTRAQ-based proteomic analysis. The overall experimental design is shown in Appendix A. Detailed information on the experimental pig population, phenotypic traits, and the characteristics of Bf and Sol were previously described by Li et al. [6]. 

### 2.2. iTRAQ Assays

#### 2.2.1. Protein Extraction

The cold acetone method was used to extract the total protein from the six muscle samples, Bf28, Bf35, Bf36, Sol28, Sol35, and Sol36. Briefly, the six muscle samples were ground into powder in liquid nitrogen, and lysis buffer, including ethylenediaminetetraacetic acid (EDTA, 2 mM) (Lingfenghx, Shanghai, China), phenylmethanesulfonyl fluoride (PMSF, 1 mM) (Beyotime, Shanghai, China), and dithiothreitol (DTT, 10 mM) (Promega, Madison, WI, USA), was added and mixed thoroughly, then centrifuged at 25,000× *g* for 20 min at 4 °C to collect the supernatant. Next, a 5-fold volume of cold acetone with 10 mM DTT was added to the supernatant and incubated at −20 °C overnight and centrifugated at 25,000× *g* at 4 °C for 20 min to collect the pellets. Then, 1.5 mL of cold acetone and 10 mM DDT were added to the samples, followed by centrifugation at 25,000× *g* at 4 °C for 20 min to collect the pellets, which were dried in the air. 

Subsequently, each pellet was resuspended using 1 mL of protein extraction reagent [8 M urea, 4% (*w*/*v*) CHAPS (Roche, Basle, Switzerland), 30 mM HEPES (Sigma, Fremont, CA, USA), 1mM PMSF, 2 mM EDTA, and 10 mM DTT], sonicated for 5 min, then centrifuged at 25,000 × g for 20 min at 4 °C, to collect the supernatant. The supernatant was added to 10 mM DDT (final concentration) and incubated at 56 °C for an hour, to reduce the disulfide bonds in proteins, before 55 mM iodoacetamide (IAM, final concentration) (Sigma, Fremont, CA, USA) was added in a dark room for 45 min to block the cysteine. Finally, 5-fold volume cold acetone with 10 mM DTT was added to the pellets, followed by incubation at −20 °C overnight and centrifugation at 25,000× *g* at 4 °C for 20 min to collect the proteins, which were dried in air. Each pellet was resuspended in 1 mL of protein extraction reagent and sonicated for 5 min. The protein concentration and quality were determined with a 2-D Quant Kit (General Electric Company, Boston, MA, USA) and verified by sodium dodecyl sulfate–polyacrylamide gel electrophoresis (SDS-PAGE) on 12% gel. The samples were stored at −80 °C for further analysis.

#### 2.2.2. Protein Digestion and iTRAQ Labelling

Protein digestion was conducted with trypsin at a ratio of 1:20 (trypsin/protein). Briefly, a 100 μL equivalent volume of protein sample was mixed with tetraethylammonium bicarbonate (TEAB) at pH 8.5 and digested at 37 °C for 4 h, followed by treatment with trypsin digestion solution, again at 37 °C for 8 h. The solvent was removed using a SpeedVac vacuum concentrator (Thermo Fisher Scientific, Waltham, MA, USA).

Subsequently, iTRAQ labelling was performed using an iTRAQ labelling kit (Applied Biosystems, Foster, CA, USA) following the manufacture’s protocol. Briefly, for each protein sample, 100 μg of protein was denatured and the cysteines were blocked, then digested with 5 μg of sequencing-grade modified trypsin (Promega, Madison, WI, USA) at 37 °C for 36 h. The trypsin-digested samples were analyzed via MALDI-TOF-TOF to ensure complete digestion and dried in a centrifugal vacuum concentrator. Following this, the protein pellets were dissolved in 30 μL of 50% TEAB (Sigma, Fremont, CA, USA) together with 70 μL of isopropanol and labelled with the 8-plex iTRAQ reagent. The protein samples were labelled with iTRAQ tags, as follows: 114 (Bf28), 116 (Bf35), 118 (Bf36), 115 (Sol28), 117 (Sol35), and 119 (Sol36). Then, the iTRAQ-labelled protein samples were pooled and subjected to strong cation exchange (SCX) fractionation.

#### 2.2.3. Strong Cation Exchange Fractionation

SCX was conducted on a high-performance liquid chromatography (HPLC) system (LC-20AB, Shimadzu, Tokyo, Japan) with an SCX column (Ultremex column, 4.6 mm I.D. × 250 mm, Phenomenex, Torrance, CA, USA). The retained peptides were dissolved using 4 mL buffer A (25 mM NaH_2_PO_4_ in 25% ACN, pH 2.7), and eluted using Buffer A for 10 min, 5–35% Buffer B (25 mM NaH2PO4, 1M KCl in 25% ACN, pH2.7) for 20 min, and 35–80% buffer B for 1 min, with at flow rate set at 1 mL/min when peptides flowed into the columns. The fractions were collected every 15 min after sample injection and desalted using a Strata-X 33-μm Polymeric Reversed Phase column, then dried in a vacuum concentrator and dissolved with 0.1% formic acid prior to reverse-phase nano-liquid chromatography/tandem mass spectrometry (nLC-MS/MS).

#### 2.2.4. nLC-MS/MS Analysis

The nLC-MS/MS analysis was performed on a Proxeon Easy Nano-LC system (Thermo Fisher Scientific, Waltham, MA, USA) connected to a hybrid quadrupole/time-of-flight MS (TOF-5600, Bruker, Leipzig, Germany). Specifically, the peptide content in each fraction was first equalized, and a 10-μL aliquot of each fraction was injected twice into the Proxeon Easy Nano-LC system. The peptides were first separated using a C18 analytical reverse-phase column with mixtures of Solution A (5% acetonitrile/0.1% formic acid) and B (95% acetonitrile/0.1% formic acid) at a flow rate of 300 nL/min. Then, the peptides were eluted using a linear LC gradient elution program, as follows: the column was equilibrated with 5% Solution B for 10 min; then, the following linear LC gradient elution procedure was used for peptide elution: 5–45% Solution B from 10 to 80 min; 45–80% Solution B from 80 to 85 min and 80% Solution B maintained for 15 min; 80–5% Solution B from 80 to 105 min and 5% Solution B held for 15 min. The SCX peptide fractions were pooled together to obtain 17 fractions, to reduce the peptide complexity, and detected using a hybrid quadrupole/time-of-flight MS (TOF-5600, Bruker, Leipzig, Germany) equipped with a nanoelectrospray ion source. All of the mass spectrometry data were collected using a Bruker Daltonics micrOTOFcontrol and processed and analyzed using data analysis software (Bruker Daltonics, Bremen, Germany). The MS/MS scans were recorded from 50 to 2000 *m*/*z*. Nitrogen was used as the collision gas. The ionization tip voltage and interface temperature were set at 1250 V and 150 °C, respectively.

### 2.3. iTRAQ Data Processing and Analysis

#### 2.3.1. Identification of Proteins and Analysis

The collected raw files were converted into MGF files for the identification of proteins. The UniProt databases were downloaded and integrated into the Mascot search engine (version 2.3.02, London, UK). The parameters for the protein identification were set as follows: trypsin was specified as the digestion enzyme, cysteine carbamidomethylation as a fixed modification, iTRAQ 8Plex on the N-terminal residue, iTRAQ 8Plex on tyrosine (Y), iTRAQ 8Plex on lysine (K), glutamine as pyroglutamic acid, and oxidation on methionine (M) as a variable modification. The tolerance settings for peptide identification in the Mascot searches were 0.05 Da for MS and 0.05 Da for MS/MS. The Mascot search results were exported into a DAT FILE and normalized and quantified using Scaffold version 3.0 software (version Scaffold_4.7.2, Proteome Software Inc., Portland, OR, USA). Confident protein identification is based on at least one unique peptide with an FDR <0.01. Protein quantification was based on the intensity of the reported ions of the assigned peptides with at least two unique spectra, and Bf28 was used as a reference to calculate the relative protein abundance. Subsequently, principal component analysis (PCA) was performed to detect the differences among the muscles, using all the protein expression data. Furthermore, gene ontology (GO) annotation and classification for all the proteins were conducted using the WEGO (web Gene Ontology Annotation Plot) web service (https://biodb.swu.edu.cn/cgi-bin/wego/index.pl, 4 May 2022). Annotation and classification of clusters of orthologous groups of proteins (KOG) for all proteins were performed according to the COG database (http://www.ncbi.nlm.nih.gov/COG/, 4 May 2022). Kyoto Encyclopedia of Genes and Genomes (KEGG) pathway annotation was conducted using the KEGG Pathway Database (http://www.genome.jp/kegg, 4 May 2022).

#### 2.3.2. Identification of Differentially Abundant Proteins (DAPs) and Analysis

DAPs were determined using Fisher’s test, according to the following criteria: fold change ≥1.2 or ≤0.83 and a *p*-value < 0.05. Hierarchical clustering of protein expression for DAPs was conducted using the online tool Pretty Heatmap, based on the Pearson distance matrix method (http://www.ehbio.com/, 20 July 2022). Moreover, eleven DPAs were randomly selected, and the corresponding differences in mRNA expression between Bf and Sol muscles were detected using qRT-PCR. The primers for DAPs detection are shown in Appendix A. The relative gene expression was calculated using the 2^−^^△△Ct^ method, and *HPRT* was used as an endogenous reference gene. Furthermore, GO enrichment analysis for DAPs was conducted using Blast2GO (http://www.blast2go.org, 4 May 2022), and KEGG pathway enrichment analysis was conducted using the KEGG Pathway Database (http://www.genome.jp/kegg, 4 May 2022). The hypergeometric test was used to calculate the *p*-values, and *p*-values < 0.05 were determined to be significantly enriched. Protein–protein interaction (PPI) analysis for all the DAPs, and the analysis of DAPs from the top 20 significantly enriched KEGG pathways were carried out using the web tool STRING with a medium confidence value of 0.4 and four active interaction sources: co-occurrence, experimental evidence, existing databases, and text-mining. The protein networks were clustered by the MCL algorithm with an inflation parameter of 3 (http://string-db.org/, 20 July 2022).

### 2.4. Integrative Analysis of the Proteome and Transcriptome Data

Previously, we constructed six transcriptome libraries, using *Biceps femoris* (Bf28, Bf35, Bf36) and *Soleus* (Sol28, Sol35, and Sol36), and carried out gene expression analysis and the identification of differentially expressed genes (DEGs) [6]. Here, we performed a correlation analysis of all the gene expressions in the transcriptome and proteome data. First, the ratio of Sol to Bf was calculated using the expression of all genes and proteins in the transcriptome and proteome data and the expression ratio was transformed to log_2_ (expression ratio). Then, the Pearson correlation between them was analyzed for all the genes and proteins identified, and the expression association of genes and proteins was shown using a nine-quadrant diagram. Furthermore, we re-identified the DEGs using the R packages DEseq2 (version 3.15), based the pig reference genome sequence (*Sscrofa* 11.1) with the criteria of “FDR < 0.05 and absolute value of log_2_ (fold change) > 1”. Then, we performed intersection analysis using the DEGs and the genes corresponding to the DAPs.

## 3. Results

### 3.1. Summary of iTRAQ-Based Proteome Data

After LC-MS/MS analysis, a total of 375,816 spectra were generated, which included 32,593 matched spectra, corresponding to 19,091 unique spectra, 4417 peptides, and 3769 unique peptides. Ultimately 1002 proteins (Appendix A) were identified in all skeletal muscle samples according to the criterion described in the “Materials and Methods”. The peptide length, protein mass, distribution of protein sequence coverage, and peptide number are shown in Appendix A. The distribution of peptide length was close to a normal distribution. Among the identified proteins, about 63.3% of the proteins had molecular weights ranging from 20 to 60 kD. Moreover, the number of proteins decreased with the increase in protein sequence coverage. Most of the proteins contained fewer than 10 peptides, and the number of proteins also decreased with the increase in peptides.

### 3.2. Functional Annotation and Classification of the Identified Proteins

In order to elucidate the biological significance of proteins, GO and KOG annotation and classification, and KEGG pathway annotation were first performed using all proteins identified in the skeletal muscle. The results are shown in Figure 1 and Appendix A. 

As the GO analysis shows in Figure 1A, the function of proteins was mainly categorized as “cellular process” and “metabolic process” for “biological process”; “cell” and “cell part” for “cellular component”; and “biding” and “catalytic activity” for “molecular function”. As the COG analysis shows in Figure 1B, most of the proteins were classified as “signal transduction mechanisms”, followed by “posttranslational modification, protein turnover, and chaperones”. Moreover, relatively more proteins were classified as “cytoskeleton”, “intracellular trafficking, secretion, and vesicular transport”, and “energy production and conversion”. As the KEGG pathway annotation shows in Appendix A, many proteins were classified in pathways related to skeletal muscle metabolism, such as “oxidative phosphorylation”, “carbon metabolism”, “glycolysis/gluconeogenesis”, “biosynthesis of amino acids”, “fatty acid degradation”, “pyruvate metabolism”, and “citrate cycle (TCA cycle)”. Moreover, many proteins were enriched in pathways associated with human diseases, such as “Parkinson’s disease”, “Alzheimer’s disease”, and “Huntington’s disease”. Overall, the identified proteins were closely related to the physiological and metabolic properties of skeletal muscle.

### 3.3. DAP Identification and GO Terms and KEGG Pathways Enrichment Analysis

After the identification of proteins, the relative protein abundance was calculated and a PCA was carried out using the expression level of all the identified proteins. The PCA results showed that there were obvious differences between the Bf and Sol samples (Figure 2), suggesting that the reliable DAPs could be obtained using Bf and Sol. 

According to the criteria of a fold change ≥1.2 or ≤0.83 and a *p*-value < 0.05, 126 DAPs between Bf and Sol were identified in this study, including 38 downregulated DAPs and 88 upregulated DAPs (Figure 3, Appendix A). 

The cluster analysis based on the protein abundance of DAPs showed that the three biological replicates of Bf or Sol clustered into one group (Appendix A), suggesting that the reliable DAPs were obtained between fast- and slow-twitch muscles. In addition, the reliability of DAPs was validated at the transcriptional level using qRT-PCR. The results showed a high consistency between the expression of proteins and mRNAs in our study, and the Pearson correlation coefficient of the log_2_ (fold change) data between the qRT-PCR and iTRAQ was 0.84, although 2 of 11 proteins showed the opposite expression patterns (Appendix A). Subsequently, GO and KEGG pathway enrichment analyses were conducted to explore the functions of the DAPs (Figure 4 and Appendix A). 

The GO analysis results showed that “mitochondrion”, “transmembrane transporter activity”, and “transmembrane transport” were the most significantly enriched GO terms for “cellular component”, “molecular function”, and “biological process”, respectively. Moreover, “organelle”, “catalytic activity”, and “single-organism process” were the most significantly enriched GO terms for “cellular component”, “molecular function”, and “biological process”, respectively. The KEGG pathway analysis results showed that the metabolism-related pathways were dominated by the significantly enriched pathways, and “cardiac muscle contraction” and “fatty acid metabolism” were the two most significantly enriched pathways. Moreover, some skeletal muscle fiber-related signaling pathways were significantly enriched, such as the “calcium signaling pathway” and the “cGMP-PKG signaling pathway”.

### 3.4. Interaction Network of DAPs

Protein interactions were analyzed for all the DAPs (Figure 5A), and the DAPs from the top 20 most significantly enriched KEGG pathways (Figure 5B) were determined using the website STRING. 

The results obtained from all the DAPs showed that the PPI enrichment *p*-value was lower than 1.0 × 10^−16^ and the average local clustering coefficient was 0.54, indicating that these DAPs had significant interaction connections. The cluster analysis results showed that all the DAPs were clustered into 18 groups, among which the average local clustering coefficient of the top five clusters ranged from 0.7 to 0.9, indicating that the DAPs in these pathways had a strong interaction. The first cluster had the largest number of DAPs (17), mainly involving the skeletal-muscle-associated structural proteins, such as MYH2, MYH7, MYL3, ACTN2, TNNC1, and TNNT1. The DAPs in the second cluster were mainly involved in the mitochondria and energy metabolism, such as COX5A, COX6C, COX7A1, ATP5L, ATP5PD, ATP5J2, and ATP5B. The DAPs in the third cluster were mainly involved in the tricarboxylic acid cycle. The DAPs in the fourth cluster were mainly involved in the fatty acid metabolism. The DAPs in the fifth cluster were related to kinase activity. Similarly, the results from the DAPs in the top 20 most significantly enriched KEGG pathways showed that the PPI enrichment *p*-value was lower than 1.0 × 10^−16^ and the average local clustering coefficient was 0.66, indicating that these DAPs had significant interaction connection. The cluster analysis results showed that the DAPs were clustered into seven groups, and the DAPs in the first two clusters were mainly involved in mitochondria and energy metabolism.

### 3.5. Association Analysis between Proteome and Transcriptome

The expression of genes in the transcription and translation levels is often inconsistent. To explore the association between the proteome and transcriptome, we calculated the Pearson correlation coefficient using the log_2_ (expression ratio) of all the genes and proteins, and the expression patterns of genes in transcription and translation levels are shown using a nine-quadrant diagram (Figure 6A). 

The results showed that there was a high association, with a Pearson correlation coefficient of 0.63 between gene expression in the proteome and transcriptome in the present study, indicating a good consistency between the proteome and transcriptome in our study. The detailed annotation information of genes in each quadrant is shown in Appendix A. Moreover, a total of 295 DEGs with the “FDR < 0.05 and absolute value of log_2_ (fold change) > 1” were identified, and 87 no redundant genes corresponding to 126 DAPs were extracted. A total of 12 genes overlapped between DEGs and DAPs (Figure 6B). These genes include *MYBPC1* (ENSSSCG00000000866), *MYH7* (ENSSSCG00000002029), *ACTN2* (ENSSSCG00000010144), *ANKRD2* (ENSSSCG00000010522), *MYL3* (ENSSSCG00000011325), *TNNC1* (ENSSSCG00000011441), *LMCD1* (ENSSSCG00000011538), *CSRP3* (ENSSSCG00000013354), *TNNT1* (ENSSSCG00000025353), *TNNI1* (ENSSSCG00000024061), *HSPB6* (ENSSSCG00000023498), and a novel gene (ENSSSCG00000039506), which represent the critical candidate genes for the formation of skeletal muscle fibers. 

## 4. Discussion

It had been demonstrated that the difference in skeletal muscle fiber types is one of the critical factors affecting meat quality [23,24,25,26]. The identification of the critical genes affecting the skeletal muscle fiber types and elucidating their regulatory mechanisms will be of great significance for the improvement of meat quality traits in livestock. Bf and Sol are two types of skeletal muscle tissues with significant differences in meat color and the expression of muscle-fiber-type-associated marker genes in pigs [6]. Previously, we constructed whole-transcriptome profiles using Bf and Sol; and identified many critical protein-coding genes [6], non-coding genes including microRNAs [27], long non-coding RNAs (lncRNAs) [8], and circular RNAs (circRNAs) [7] related to skeletal muscle fibers. It is well known that these proteins are the end product of coding genes executing a function, and these expressions of genes may be inconsistent at the transcriptional or translational levels. Thus, the aim of this study was to screen the DAPs between Bf and Sol using the proteomic technique and combine the transcriptome data to find the critical genes related to porcine skeletal muscle fibers and pork quality. 

The iTRAQ-based proteomic method is a technique that is commonly used for protein relative quantification [28]. In the present study, 1002 proteins were identified. The number identified was relatively small compared to the number identified in some studies using iTRAQ assays [22,29,30] but is consistent with the proteomic research of skeletal muscle in pig [20]. These different results may be influenced by species, tissues, and the process used for the detection of proteins. Moreover, the correlation analysis showed a good consistency between the observed genes and the expression of proteins (Figure 6A), which is consistent with other proteomic studies conducted using iTRAQ assays [31]. Additionally, the consistency between the genes and proteins expression was further validated through the quantification of DAPs using a real-time PCR method (Appendix A). However, our results are not consistent with previous studies in human and mouse [32,33], in which, although the skeletal muscle tissues were used, an overall low correlation between mRNA and protein was observed. These differences may be influenced by species and physiological status.

According to the criteria commonly adopted (fold change ≥1.2 or ≤0.83 and a *p*-value < 0.05), we identified 126 DAPs between Bf vs. Sol, which represent the critical candidate proteins affecting the characteristics of skeletal muscles. Many DAPs are involved in the structure of skeletal muscle, such as PVALB, MYH7, MYL2, MYL3, TNNC1, MYH2, TNNI1, and TNNT1, and MYH7 and MYH2 are involved in skeletal muscle fiber types, [34,35], suggesting the DAPs identified in this study are promising candidate genes affecting meat quality. Notably, PVALB is related to Ca^+^ metabolism [36] and was reported to be a critical candidate gene affecting pork quality [37]. Other DAPs are associated with the mitochondria and energy metabolism of muscle, such as ATP5B, ATP2A1, PKM, COQ6, COX6C, CA3, AK1, and HADHA. This can also be seen in the results of protein interactions (Figure 6). Intriguingly, the PHKG1 and RYR1 have been proven to be two major genes affecting pork quality traits [38,39] and were also the DAPs in this study, supporting the importance of the DAPs identified here. Carbonic anhydrase 3 (CA3) is a key enzyme mediating the reversible hydration of carbon dioxide, is enriched in slow-twitching type I fibers, and can be used as an ideal marker for studying fiber-type shifting and muscle adaptations [40]. In addition, some DAPs have been demonstrated to affect meat quality, such as LDHB [41], AMPD1 [42], and PKM [43]. Moreover, in the KEGG enrichment analysis, skeletal-muscle-fiber-related signaling pathways, such as the “Calcium signaling pathway (ko04020)”, were found to be significantly enriched. Additionally, “Fatty acid metabolism (ko01212)”, “Fatty acid degradation (ko00071)”, and “Fatty acid elongation (ko00062)” were also found to be significantly enriched, suggesting that the skeletal muscle fiber types may be impacted by fatty acid metabolism. Generally, many of the KEGG pathways enriched in proteomic analysis in the present study overlapped with those in our previous transcriptome analysis [6,7,8,27], indicating that the overlapped enriched KEGG pathways may play a crucial role in the formation of skeletal muscle fibers. Finally, we screened 12 genes between DEGs and DAPs in an overlapping analysis; the pathways involved in these genes are critical for the formation of skeletal muscle fibers. 

## 5. Conclusions

In summary, we conducted the differential proteomic profiling of fast- and slow-twitch muscles using an iTRAQ-based proteomic method and identified 126 DAPs between Bf and Sol, deepening our understanding of the difference between fast- and slow-twitch muscles at the protein level. Notably, after a combined analysis with transcriptome profiling data, 12 genes were found to overlap between DEGs and DAPs, which are the key candidate genes regulating the formation of skeletal muscle fibers. Moreover, the results obtained from the functional classification, the enrichment analysis of DAPs, and the PPI network provide valuable information for the regulation of skeletal muscle fiber. Overall, our study reveals new promising key proteins controlling skeletal muscle fiber type formation, which may be useful for the improvement of meat quality traits in pigs. Further studies are still needed to reveal the roles of these DAPs, the enriched pathways, and the PPI network in the regulation of skeletal muscle fiber types. 

## Figures and Tables

**Figure 1 foods-11-02842-f001:**
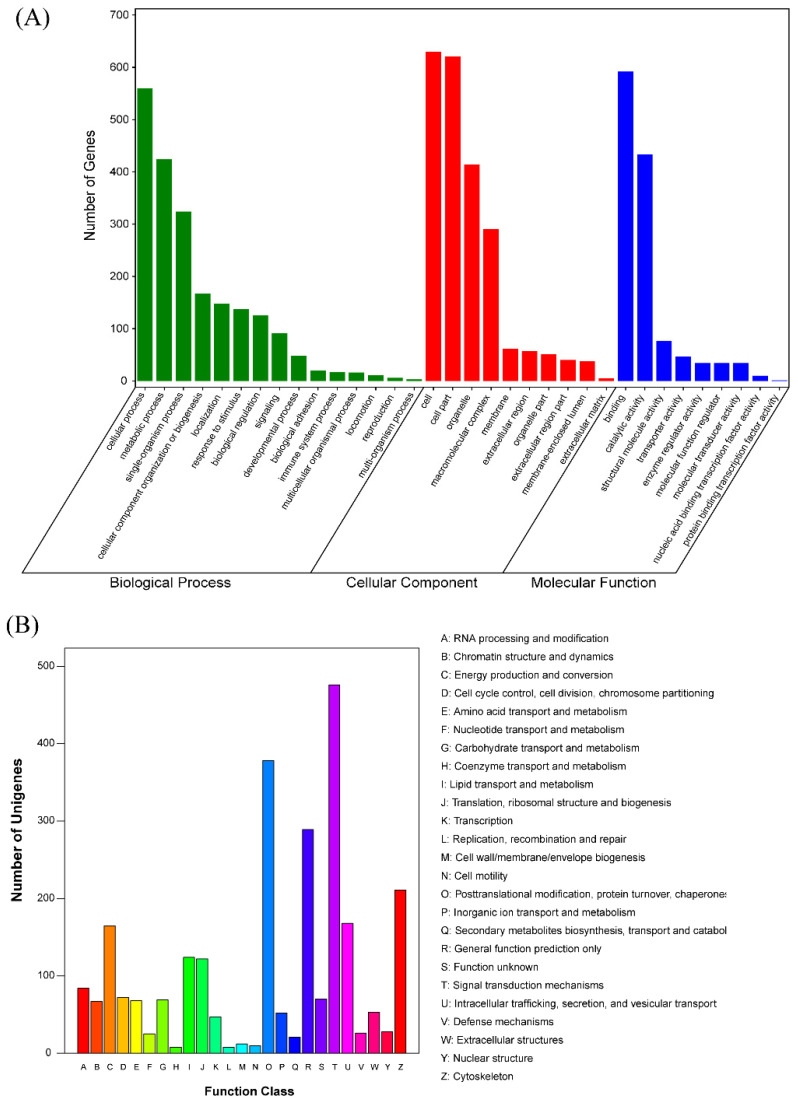
GO and KOG annotation and classification of all identified proteins. (**A**) GO classification of all the proteins identified. *x* axis represents the GO terms in three types of GO ontology, and the *y* axis represents the gene number. (**B**) KOG classification of all proteins identified. *x* axis refers to function class; *y* axis represents the gene number.

**Figure 2 foods-11-02842-f002:**
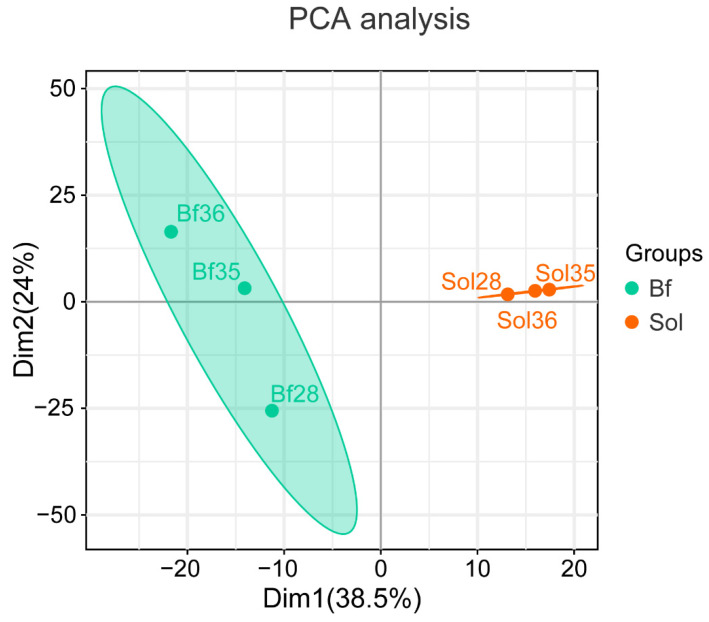
Principal component analysis (PCA). Based on the expression of the identified proteins, PCA was carried out to visualize the differences between Bf and Sol samples.

**Figure 3 foods-11-02842-f003:**
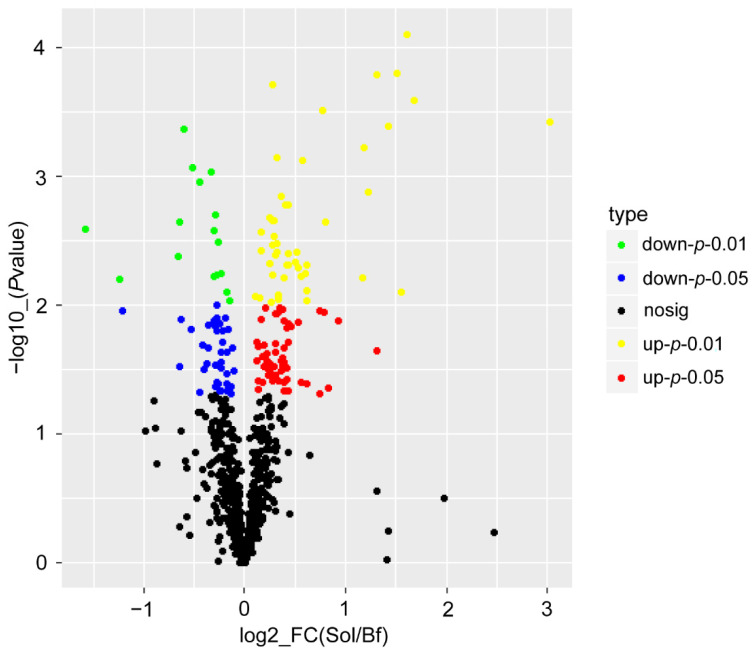
Identification of DAPs between fast-twitch and slow-twitch muscles. A volcano plot was drawn to show the DAPs between Bf and Sol. The green dots indicate significantly downregulated proteins (*p* < 0.01 and fold change <0.83), the blue dots indicate significantly downregulated proteins (*p* < 0.05 and fold change <0.83), the yellow dots indicate significantly upregulated proteins (*p* < 0.01 and fold change >1.2), the red dots indicate significantly upregulated proteins (*p* < 0.05 and fold change >1.2), and the black dots represent proteins with non-significant (*p* > 0.05 or 0.83 < fold change <1.2) differences in expression.

**Figure 4 foods-11-02842-f004:**
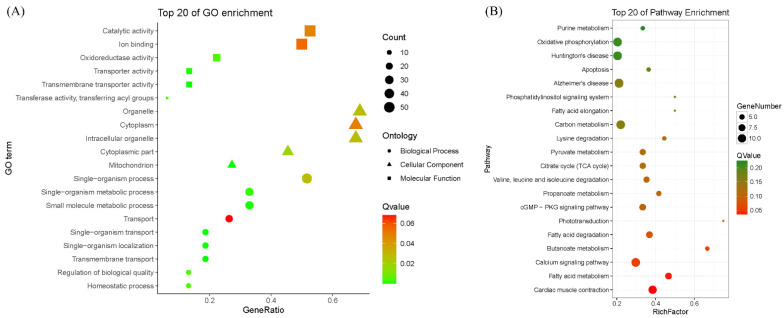
GO and KEGG pathway enrichment analysis of DAPs. (**A**) GO enrichment analysis of DAPs. (**B**) KEGG pathway enrichment of DAPs. The top 20 enrichment GO terms and pathways are shown, and detailed information can be found in Appendix A.

**Figure 5 foods-11-02842-f005:**
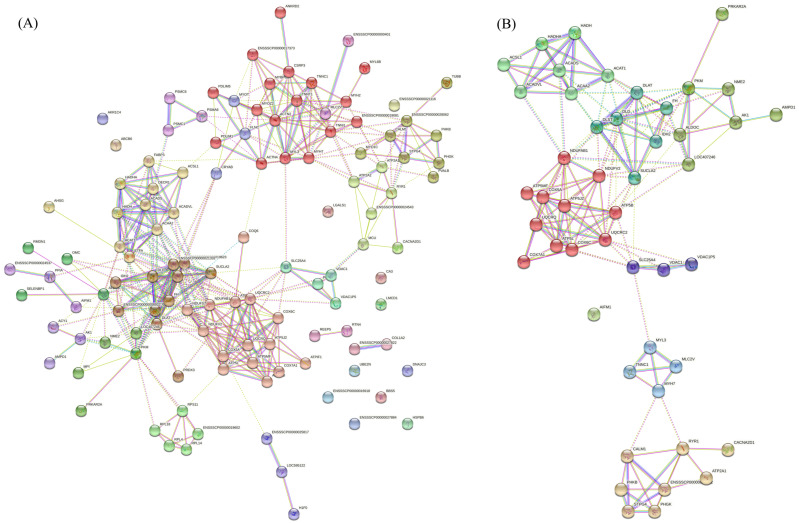
Protein–protein interaction (PPI) network of DAPs. (**A**) PPI network of all the DAPs. (**B**) PPI network of the DAPs from the top 20 most significantly enriched KEGG pathways. The interaction network was constructed using the web-based search STRING database. Line color indicates the type of interaction evidence: the light blue line indicates the known interactions from the curated databases, the purple line indicates the known interactions that were experimentally determined, the blue line indicates the predicted interactions of co-occurrence genes, and the yellow line indicates the PPI using the text mining method. Solid line represents the PPIs that were experimentally validated, whereas the dotted line represents the PPIs that have not yet been validated.

**Figure 6 foods-11-02842-f006:**
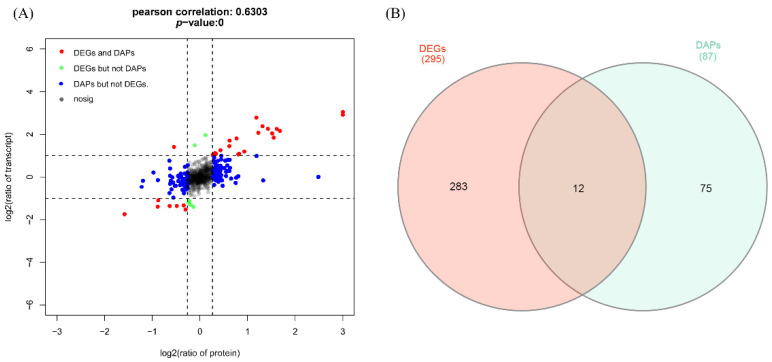
Integrated analysis of transcriptome and proteome. (**A**) Overall correlation between mRNA and protein changes displayed using a nine-quadrant diagram. *x* axis represents log_2_ (fold change) of proteins and *y* axis represents the log_2_ (fold change) of mRNAs. The correlation coefficient and *p* value of the transcriptome and proteome are shown at the top of the graph. Each dot represents a gene and protein. The black dots represent non-differentially expressed proteins and genes; the red dots represent the genes and proteins whose expression trends are consistent or opposite; the green dots represent the DEGs but not DAPs; and the blue dots represent DAPs but not DEGs. (**B**) Venn plot of DAPs and DEGs. The gene name and Ensemble ID for the 12 overlapped genes are shown in the “Results” section of the main text.

## Data Availability

Data is contained within the article and Appendix A.

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
