# Peer review of "A Combined Differential Proteome and Transcriptome Profiling of Fast- and Slow-Twitch Skeletal Muscle in Pigs"

_foods, 2022, doi:10.3390/foods11182842_

Round 1
Reviewer 1 Report
Abstract
Lines 16-18: I believe the previous findings should not be mentioned in the abstract; please remove them.
Line 75 : the reference (mach et al,,,,,,,)year is not included or the number as the journal required ?
Line 111: please add the six ……..
Line 210: the detection primers,,,,,,,, the paragraph should be rephrased in corrected language
The figures should be modified to be larger and readable
More discussion is needed
Line 431 , in conclusion language errors or typing errors
Number of samples used may affect the results (low number)
The language should be corrected
Reviewer 2 Report
The manuscript reports the results of a proteomic study focused on pig muscles. Many parts need to be substantially revised and improved and re-addressed properly. The manuscript needs major revisions in many parts.
English needs to be improved in many parts
The title: it does not reflect the content of the work
This is not a genetic study therefore it does not provide any information on the genetic basis of skeletal muscle fibers in pigs - and then what does it mean ... basis of skeletal muscle fibers ...?
This is wrong concept is also reported in the conclusions - this part should be substantially re-evaluated and redifined
Abstract: It should be improved substantially. For example, the following first sentence:
"The difference of skeletal muscle fiber types is the main cause affecting the pork quality." is not true - it could be eventually:
"Skeletal muscle fiber types can contribute in part to affect pork quality parameters."
This concept should be adjusted in the introduction and in all other parts of the text
Introduction: Sentences are too long. Introduction should be also revised substantially in the content. Several sentences are not correct or are misleading
lines 44-50: these sentences do not make any sense and are irrelevant in this context- please eliminate - eliminate these elements also from the discussion
lines 51-52 - this sentence is not correct
lines 52-54: this sentence does not make any sense
Line 72: house?
Line 82: genetic basis ? No, this is a wrong concept
Line 85-86: wrong concept, please see above
M&M
The histological characterization of the sampled muscle is missed - therefore there is a substantial problem here - the fiber types are not defined, the assumption on the different fiber types is based on the two muscles that are used for sampling
Round 2
Reviewer 2 Report
The manuscript has been improved